# Development of a confinable gene drive system in the human disease vector *Aedes aegypti*

**Ming Li[1], Ting Yang[1], Nikolay P Kandul[1], Michelle Bui[1], Stephanie Gamez[1], Robyn Raban[1], Jared Bennett[2], Héctor M Sánchez C[3], Gregory C Lanzaro[4], Hanno Schmidt[4], Yoosook Lee[4], John M Marshall[3,5], Omar S Akbari[1,6]\***

[1]Section of Cell and Developmental Biology, University of California, San Diego, San Diego, United States; [2]Department of Biophysics, University of California, Berkeley, Berkeley, United States; [3]Division of Epidemiology and Biostatistics, School of Public Health, University of California, Berkeley, Berkeley, United States; [4]Vector Genetics Laboratory, Department of Pathology, Microbiology, and Immunology, School of Veterinary Medicine, University of California, Davis, Davis, United States; [5]Innovative Genomics Institute, Berkeley, United States; [6]Tata Institute for Genetics and Society, University of California, San Diego, La Jolla, United States

**Abstract** *Aedes aegypti* is the principal mosquito vector for many arboviruses that increasingly infect millions of people every year. With an escalating burden of infections and the relative failure of traditional control methods, the development of innovative control measures has become of paramount importance. The use of gene drives has sparked significant enthusiasm for genetic control of mosquitoes; however, no such system has been developed in *Ae. aegypti*. To fill this void, here we develop several CRISPR-based split gene drives for use in this vector. With cleavage rates up to 100% and transmission rates as high as 94%, mathematical models predict that these systems could spread anti-pathogen effector genes into wild populations in a safe, confinable and reversible manner appropriate for field trials and effective for controlling disease. These findings could expedite the development of effector-linked gene drives that could safely control wild populations of *Ae. aegypti* to combat local pathogen transmission.

**\*For correspondence:**
oakbari@ucsd.edu

**Competing interests:** The authors declare that no competing interests exist.

## Introduction

Due to the high annual incidence of vector-borne disease, the associated economic burdens, and the lack of effective vaccines, interest in the innovation of novel population-control methods to prevent pathogen transmission is increasing. *Aedes aegypti* is the principal mosquito vector of several arboviruses, including yellow fever, chikungunya, dengue, and Zika (*Scott and Takken, 2012*). This mosquito alone places roughly half of the world's population at risk of acquiring vector-borne diseases while causing an estimated 390 million dengue infections annually (*Bhatt et al., 2013*). Traditional mosquito control strategies, including long-lasting insecticide-treated bed nets (LLINs), chemical insecticides, and environmental management (*Schreck, 1991*), have significantly reduced mosquito-borne disease burdens; however these interventions are largely unsustainable due to the development of both genetic and behavioral vector resistance that decrease their efficacy (*Vontas et al., 2012*), high costs, and the need for repeated applications with limited spatial impact. Furthermore, chemical interventions, a fundamental tool for vector control, negatively affect non-target beneficial organisms, such as pollinators. Therefore, there is a pressing global need for alternative safe and effective approaches to control mosquito disease vectors.

Species-specific mosquito population suppression technologies, such as the *Wolbachia* incompatible insect technique (IIT) and the release of dominant lethal gene (RIDL), are currently being implemented on a small-scale in the field to control populations of *Ae. aegypti* (*Harris et al., 2012*; *McMeniman et al., 2009*; *Phuc et al., 2007*). While these technologies improve *Ae. aegypti* control capabilities, they require continuous inundation, which is laborious and cost prohibitive to many areas with the largest vector-borne disease burdens. In addition to IIT, *Wolbachia* has also been used for population replacement in various locations (*Hoffmann et al., 2011*; *Jiggins, 2017*; *Schmidt et al., 2017*), as certain strains can spread to fixation while reducing the vector's susceptibility to dengue virus (*Walker et al., 2011*). However, the literature suggests the effectiveness of different *Wolbachia* strains varies so far as to even enhance vector competence in some contexts, which may lead to a reevaluation of this technique (*Amuzu et al., 2018*; *Dodson et al., 2014*; *Hughes et al., 2014*; *Hughes et al., 2012*; *King et al., 2018*; *Murdock et al., 2015*; *Zélé et al., 2014*). Therefore, there has been renewed interest in developing novel species-specific and cost-effective genetic technologies that provide long-lasting disease reduction with limited effort.

One innovative technology, first articulated by *Burt (2003)*, utilizes homing-based gene-drive technologies to expedite the elimination and eradication of vector-borne diseases (*Burt, 2014*; *Champer et al., 2016*; *Esvelt et al., 2014*; *Marshall and Akbari, 2017*; *Marshall and Akbari, 2016*). Conceptually, these drives function by exploiting the organism's innate DNA repair machinery to copy or 'home' themselves into a target genomic location prior to meiosis in the germline. This process converts wild type alleles into drive alleles in heterozygotes, thereby forcing super-Mendelian inheritance of the drive into subsequent generations, irrespective of the fitness cost to the organism. Theoretically, this inheritance scheme can spread the drive and any linked anti-pathogen 'cargo genes,' to fixation in a population in a short time frame (*Buchman et al., 2019*; *Dong et al., 2018*; *Yen et al., 2018*), even with modest introduction frequencies (*Burt, 2003*). Importantly, if these drives are linked with effective anti-pathogen cargo genes, they could deliver efficient and cost-effective vector control (*Burt, 2003*). The drawback to this scheme came from the difficulty in engineering homing drives with various endonucleases, however the recent CRISPR revolution has streamlined their development (*Champer et al., 2016*; *Esvelt et al., 2014*). In fact, proof-of-concept CRISPR-based homing gene drives have recently been developed in several organisms including yeast (*Basgall et al., 2018*; *DiCarlo et al., 2015*; *Roggenkamp et al., 2018*; *Shapiro et al., 2018*), flies (*Champer et al., 2019*; *Champer et al., 2018*; *Champer et al., 2017*; *Kandul et al., 2019a*; *KaramiNejadRanjbar et al., 2018*; *Oberhofer et al., 2018*), mice (*Grunwald et al., 2019*), and two malaria vector species, *Anopheles gambiae* (*Hammond et al., 2016*; *Hammond et al., 2017*; *Kyrou et al., 2018*) and *Anopheles stephensi* (*Gantz et al., 2015*). All of these CRISPR-based homing gene drives minimally consist of a Cas9 endonuclease, which facilitates the genome integration of the gene drive and a guide RNA (gRNA) cassette, which encodes the sequence-specific integration site targeted by Cas9. When these two components are expressed together in the germline at the correct time and quantity, they can cause drive conversion at some frequency, a process in which one copy of the drive becomes two copies, thereby biasing the inheritance of the drive.

Unfortunately, a gene-drive system has yet to be developed in *Ae. aegypti*, the lack of which is principally due to the absence of tools necessary to engineer drives. In response to these deficiencies, here we molecularly characterized drive components and then evaluated and optimized these components in several drive systems in *Ae aegypti*. As a safety precaution, we engineered 'split gene drives (referred to as split drives from hereon)' with a molecularly unlinked endonuclease, thereby allowing the user to spatially and temporally confine the drive (*Akbari et al., 2015a*; *Champer et al., 2016*; *Committee on Gene Drive Research in Non-Human Organisms: Recommendations for Responsible Conduct, Board on Life Sciences, Division on Earth and Life Studies, National Academies of Sciences, Engineering, and Medicine, 2016*; *Esvelt et al., 2014*). To improve drive efficiency, we targeted a highly conserved region in a phenotypic gene with cleavage rates, indicating biallelic cutting, of 100% and homing rates, indicating the super-Mendelian transmission rate of the drive, as high as 94%. In addition to demonstrating drive efficiency, we performed mathematical modeling that suggested split drives are particularly valuable in gene drive development since: *i)* several consecutive releases of male mosquitoes harboring the split drive and anti-pathogen cargo, at an achievable 1:1 ratio with the wild population, could spread disease refractoriness into wild populations, *ii)* the split drive should not significantly spread into neighboring populations, and *iii)* dissociation of the unlinked endonuclease and guide RNA components, in

addition to the fitness costs associated with the split drive and cargo, may cause the drive to be eliminated from the population on a timescale that could accommodate local arbovirus elimination. These highly desirable features could enable safe testing of a split drive system in the field prior to the release of a more invasive, self-propagating, linked-drive system, and could lead to the development of new technologies to prevent vector-borne disease transmission.

## Results

### Rational target site selection for the drive

To develop a binary Cas9 gRNA approach that could be leveraged for the development of gene drives, we built on our previous work that generated *Ae. aegypti* strains expressing the Cas9 endonuclease in both germline and somatic cells and demonstrated their robust cleavage and improved homology directed repair (HDR) efficiencies (*Li et al., 2017*). We targeted an easily screenable gene with consistent and viable disrupted phenotype, the *white* gene (AAEL016999), which is an ATP-binding cassette (ABC) transporter (*Bhalla, 1968*; *Coates et al., 1997*; *Li et al., 2017*). To locate a highly conserved target sequence in *white*, we bioinformatically compared the whole-genome sequences of 143 *Ae. aegypti* mosquitoes sampled throughout California (n = 127), Florida (n = 4), Mexico (n = 3), Puerto Rico (n = 6), and South Africa (n = 3) (*Figure 1—figure supplement 1*, *Figure 1—source data 1*) (*Lee et al., 2019*). From this analysis, we discovered a putative gRNA target sequence in exon 3 of *white* ($gRNA^w$) that was 100% conserved in these wild populations (*Figure 1—figure supplement 2A–B*). To target this highly conserved sequence and determine cleavage efficiency at this site, we synthesized and injected gRNAs into embryos derived from *exu-Cas9* (*Exuperantia*, AAEL010097) females that maternally deposit Cas9. We achieved a 92.7 ± 5.8% mutation efficiency in the injected $G_0$s, which is consistent with our previous work in this line (*Li et al., 2017*), demonstrating that this target site is accessible to Cas9 cleavage and that this gRNA is highly active.

### Characterization of functional polymerase-III promoters

Next, we needed a promoter to ensure proper strength and timing of gRNA expression for cleavage and homing to occur in the germline. U6 non-coding small nuclear RNA (snRNA) polymerase III promoters are ideal for gRNA expression due to their nucleus-associated transcription and unmodified 5' and 3' regions (i.e., no 5' cap or 3' polyA tail); consequently, they have been used to drive gRNA expression in various species (*Gantz et al., 2015*; *Hammond et al., 2016*; *Kondo and Ueda, 2013*). Unfortunately, U6 promoters have not been functionally characterized in *Ae. aegypti*. There are six known U6 genes in *Ae. aegypti* (*Matthews et al., 2018*), however their promoters remain to be characterized. Therefore, to characterize these promoters for gRNA expression, we engineered six constructs each with a predicted U6 promoter (termed U6a-U6f from hereon) derived from *Ae. aegypti* to drive expression of $gRNA^w$ (*Figure 1—figure supplement 2C*). To transiently assess promoter activity, we microinjected each construct separately into *exu-Cas9* derived embryos (*Li et al., 2017*) (*Figure 1A*). Of the six injected constructs, only four (U6a-d) induced somatic loss-of-function variegated biallelic white-eye loss-of-function phenotypes in $G_0$ embryos with moderate mutation frequencies ranging from 32.5–71.3% (*Figure 1B*), indicating that these four promoters can transiently express $gRNA^w$. We then evaluated germline mutation rates by outcrossing mosaic $G_0$ mosquitoes to recessive *white* mutants ($w^{\Delta 19}/w^{\Delta 19}$) harboring a 19 bp deletion at the *white* gRNA cut site which would lead to heritable the recessive loss-of-function phenotypes in successfully edited mosquitoes. These phenotypes were then scored in the G1 progeny with rates ranging 27.6–67.5%, indicating efficient germline editing (*Figure 1A–B*).

### Development of a binary CRISPR approach

To measure cleavage and mutagenesis efficiencies of genome-integrated drive components, we developed a simple fluorescence-based system that would be compatible and orthogonal to the white mosaic readout. We developed this system by genetically encoding the four functional U6a-U6d-$gRNA^w$ transgenes by injecting these constructs into *Ae. aegypti* embryos (designated as wild type [*wt*]) and establishing a transgenic line for each construct. Transgenesis was confirmed by the enhanced green fluorescent protein (eGFP) transgenesis marker (*Figure 1—figure supplement 2C*), and positives were outcrossed to *wt* strains to establish stocks. Somatic cleavage efficiencies were

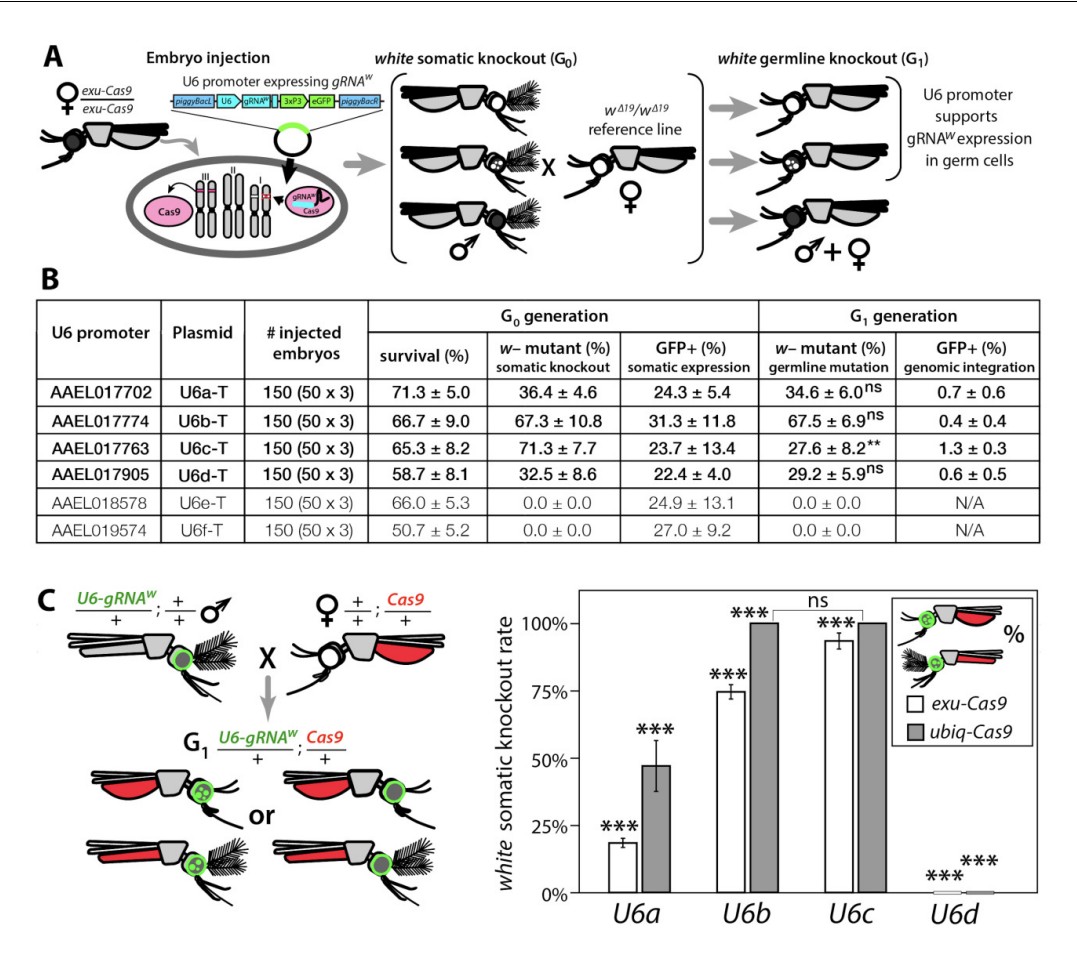

**Figure 1.** Functional identification of polymerase III promoters in *Ae. aegypti*. (**A**) *exu-Cas9* embryos were injected with one of 6 *piggyBac* plasmids, each utilizing a different U6 promoter expressing a guide RNA targeting the *white* eye pigmentation gene, gRNA^w (*Figure 1—figure supplement 2*). The frequency of somatic *white* eye phenotypes in the resulting $G_0$ progeny was used to assess promoter efficiency and to confirm gRNA target site accessibility. Germline mutagenesis rates were assessed by crossing $G_0$ to *white* loss-of-function ($w^{\Delta19}/w^{\Delta19}$) lines to determine the *white* eye phenotype frequency in $G_1$ progeny. (**B**). Two types of *w–* knockouts were observed: complete white eyes and mosaic white eyes. Out of 6 tested U6 promoters, four U6 promoters (U6a, U6b, U6c, and U6d) induced *white* knockout phenotypes. Statistical differences between germline and somatic mutation rates were estimated by equal variance *t*-test. (**C**) Transgenic males harboring *piggyBac*-integrated U6-gRNA^W were outcrossed to either *exu-Cas9* or *ubiq-Cas9* females (left panel), and eye phenotypes were scored in $G_1$ trans-heterozygous progeny (right panel, ***Supplementary file 1***). Statistical differences in mutation knockout rates were estimated by equal variance *t*-test. ($P \geq 0.05^{ns}$, $p<0.05^*$, $p<0.01^{**}$, and $p<0.001^{***}$).

The online version of this article includes the following source data and figure supplement(s) for figure 1:

**Source data 1.** The metadata of the *Aedes aegypti* whole-genome sequences.

**Figure supplement 1.** Sample locations of mosquitoes utilized for whole genome sequencing.

**Figure supplement 2.** Guide RNA (gRNA) sequence and gene constructs developed in the study.

**Figure supplement 3.** Design of the split-drive, a confined and high-threshold population replacement system.

estimated by crossing males from each line (*U6a-U6d-gRNA*^w) to females of Cas9 expression lines, *exu-Cas9* and *ubiq-Cas9* (ubiquitin L40, AAEL006511) which rely on different promoters to express Cas9 (*Li et al., 2017*). The frequency of somatic *white* mutagenesis was scored in the trans-heterozygous progeny (*Figure 1C*). Interestingly, the rates of mosaicism, an intermediate phenotype with both *wt* and knockout characteristics (i.e. black and white patchy eyes see *Figure 2*), varied depending on the Cas9 and the gRNA^w promoters. As the timing and level of both Cas9 and gRNA expression is important for inducing mosaicism (*Kandul et al., 2019b*; *Yen et al., 2014*), this could contribute to this variation, or it could also be position dependent as the integration of the transgene cassettes was not site specific. Interestingly, the combination of the *ubiq-Cas9* strain and the

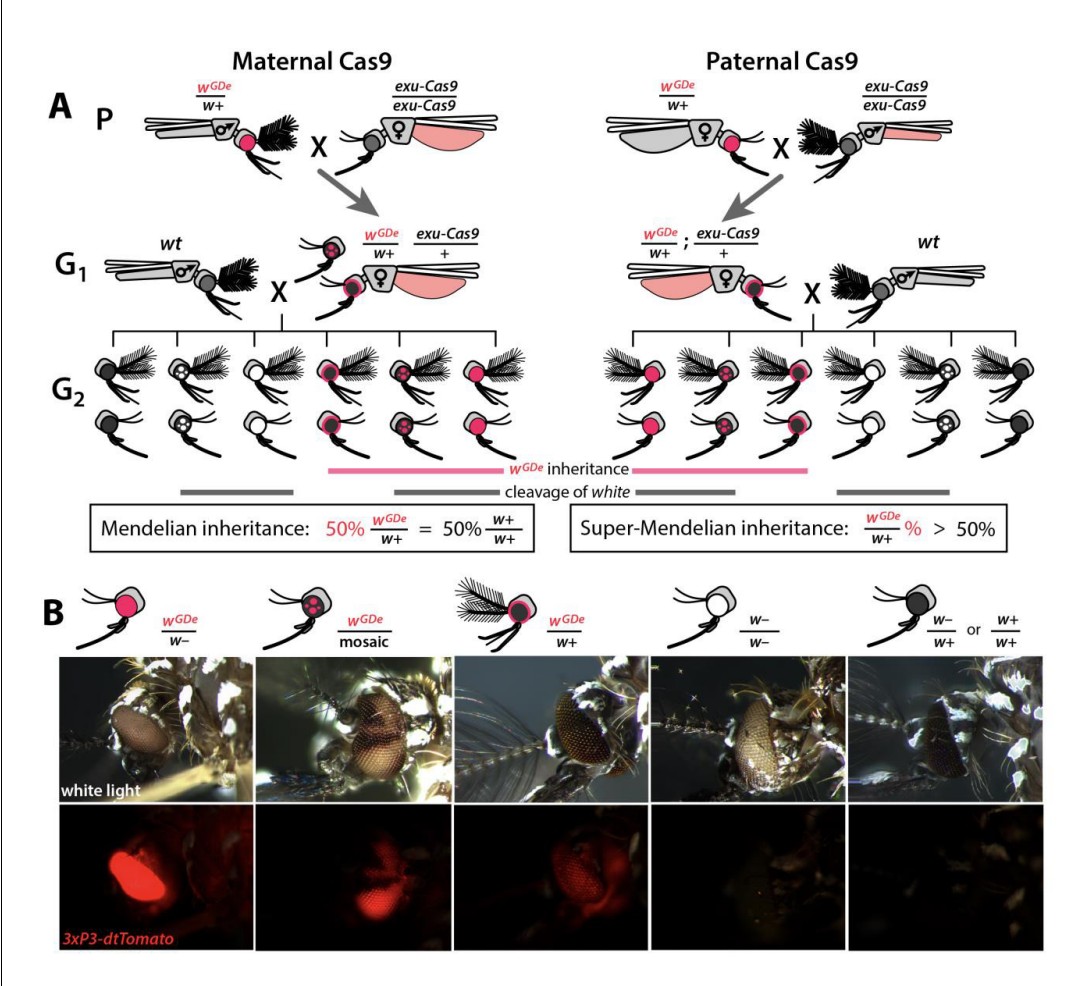

**Figure 2.** Assessing super-Mendelian inheritance of split drives. (A) Crossing scheme of the $w^{GDe}$ and *exu-Cas9* parent strains (P) to generate trans-heterozygotes ($G_1$), and the outcrossing of their progeny to wild-type (*wt*) mosquitoes ($G_2$). *tdTomato* eye and *dsRed* abdominal expression were the $w^{GDe}$ and *exu-Cas9* transgene inheritance markers, respectively. $w^{GDe}$ transmission and *white* loss-of-function mutation rates were estimated among $G_2$ progeny of trans-heterozygous female and *wt* male crosses. Super-Mendelian inheritance of $w^{GDe}$ occurred when the transmission rate of $w^{GDe}$ in $G_2$ progeny was >50%, as expected by standard Mendelian inheritance. (B) Examples of $G_2$ progeny eye phenotypes and corresponding genotypes. The online version of this article includes the following figure supplement(s) for figure 2:

**Figure supplement 1.** Sex biased inheritance of the GDe is due to linkage of *white* and *Nix*, a male determining gene in *Ae. aegypti*.
**Figure supplement 2.** Site-specific integration of Gene Drive element (GDe).

*U6b-gRNA^w* and *U6c-gRNA^w* strains induced up to 100% *white* somatic knockout frequencies, with varied rates of eye pigment mosaicism, indicating that these U6 promoters are highly active and timing appropriate when encoded in the genome to induce knock-out phenotypes (*Figure 1C*).

## Engineering split drives

Given the above promising results, we next wanted to use these components to develop split drives. To engineer split drives, we separated the drive components into two elements: a gene-drive element (GDe) and the Cas9 endonuclease ((*Figure 1—figure supplement 3*). Because we did not know which U6 promoter would be optimal for homing in the germline, we engineered four GDe's (*U6a-GDe–U6d-GDe*), each consisting of *gRNA^w* under the control of a different functional U6 promoter (U6a–U6d) that also contained a dominant *tdTomato* marker (red fluorescent protein) for tracking and two outer homology arms immediately flanking the *gRNA^w* target site to mediate HDR at the *white* locus (*Figure 1—figure supplements 2D* and *3A*). While no previous efforts have been

undertaken to improve sgRNA function in *Ae. aegypti*, to attempt to further increase gRNA expression, the gRNA scaffold was modified slightly as previously described (*Dang et al., 2015*) to eliminate cryptic termination sequences. We established transgenic lines by injecting *wt* embryos with Cas9 recombinant protein pre-mixed with *gRNA*^w (Cas9/*gRNA*^w) in combination with a GDe construct to serve as the template for site-specific integration via Cas9-mediated HDR (*Figure 2—figure supplement 1A*). Transgenic mosquitoes, $w^{GDe}/w+$, were readily identified as G$_1$ larvae by their dominant *tdTomato* expression and black (*w+*) eyes, and they were then intercrossed to establish lines. Integration of the GDe's into the *white* locus was genetically confirmed by the presence of homozygous ($w^{GDe}/w^{GDe}$) individuals expressing *tdTomato* with recessive white (*w–*) eyes, which was additionally confirmed by genomic sequencing of the left and right insertion boundaries all of which inserted perfectly except U6d-GDe which unexpectedly had an 276 bp insertion in the right homology arm (*Figure 2—figure supplement 2*).

## Sex-biased inheritance of GDe

Next, we confirmed the Mendelian inheritance of each *GDe* by reciprocal outcrossing of 20 individual heterozygous ($w^{GDe}/w+$) individuals to *wt* (*w+/w+*). The expected result was that both male and female heterozygous individuals would transmit the $w^{GDe}$ gene to their progeny at the Mendelian rate of 50%. Indeed, the heterozygous $w^{GDe}/w+$ females did meet this expectation, conferring $w^{GDe}$ inheritance to their progeny at normal Mendelian rates. The $w^{GDe}/w+$ males, however, displayed sex-segregated transmission of $w^{GDe}$ to either mostly females at 99.8 ± 0.3% (type I ♂ in *Figure 2—figure supplement 1*, *Supplementary file 2*) or to mostly males at 99.8 ± 0.2% (type II ♂ in *Figure 2—figure supplement 1*, *Supplementary file 2*). This result is not unexpected as *white* has been previously been described as a sex-linked gene in *Ae. aegypti* (*Bhalla, 1968*), and recent genome sequencing has confirmed this linkage (*Dudchenko et al., 2017*; *Matthews et al., 2018*) as *white* is located in close proximity to *Nix* (AAEL022912), a dominant male-determining factor (M-factor) located on chromosome I (*Aryan et al., 2019*; *Hall et al., 2015*). There is also little to no recombination in this region, especially in males (*Fontaine et al., 2017*; *Severson et al., 2002*; *Toups and Hahn, 2010*; *Hall et al., 2014*), which would further support sex-linkage at this loci. Therefore, depending on which chromosome the *GDe* initially integrated, the *GDe* and *Nix* were either linked, producing nearly all male progeny, which inherited both $w^{GDe}$ and *Nix* to exhibit male-biased inheritance, or the *GDe* was inherited separately from *Nix* ($w^{GDe}/w+$) to exhibit female-biased inheritance (*Figure 2—figure supplement 1*). Nevertheless, when both sexes were considered, heterozygous $w^{GDe}/w+$ parents transmitted $w^{GDe}$ to their progeny at the 50% normal Mendelian inheritance rate.

## Split drives induce super-Mendelian inheritance

To determine whether split drives could selfishly bias transmission, which could be exploited to rapidly and efficiently drive an anti-pathogen effector into a population, we performed a number of genetic crosses to determine transmission efficiencies. First, we estimated somatic cleavage efficiencies by outcrossing each $w^{GDe}$ strain to the *exu-Cas9* strain to generate trans-heterozygous $w^{GDe}/w+$; *exu-Cas9/+* individuals. We found that rates of maternal carryover-dependent cleavage, which occurs when mothers maternally deposit high levels of Cas9 into their embryos and is indicated by the presence of mosaic phenotypes, varied depending on U6 promoter (26–100%), while the paternal carryover of Cas9 was not sufficient to induce these phenotypes (*Figures 2A* and *3A*; *Supplementary file 3a*, *Supplementary file 3b*). Next, to measure homing efficiencies, we bidirectionally outcrossed G$_1$ trans-heterozygotes $w^{GDe}/w+$; *exu-Cas9/+* to *wt* and quantified somatic cleavage and inheritance frequencies in G$_2$ progeny (*Figure 2*). From these crosses, cleavage and homing efficiencies varied, most notably trans-heterozygous $w^{U6b-GDe}/w+$; *exu-Cas9/+* females induced somatic mosaicism in 96.3 ± 4.0% of their progeny (*Figure 3B*, *Supplementary file 3c*). Furthermore, modest super-Mendelian inheritance was observed (70.9 ± 7.8% in G$_2$ progeny) which is significantly greater than the expected 50% for normal Mendelian transmission, while no cleavage or homing was observed from trans-heterozygous G$_1$ male outcrosses to *wt* (*Figure 3C*; *Supplementary file 3d*). To distinguish the contribution of maternal effects to differential drive transmission, we repeated these crosses with paternal Cas9 inheritance; however, there were no significant differences in transmission rates in the G$_1$ progeny based on the parentage of the Cas9 donor (*Figures 2A* and *3B–C*; *Supplementary files 3e – 3f*).

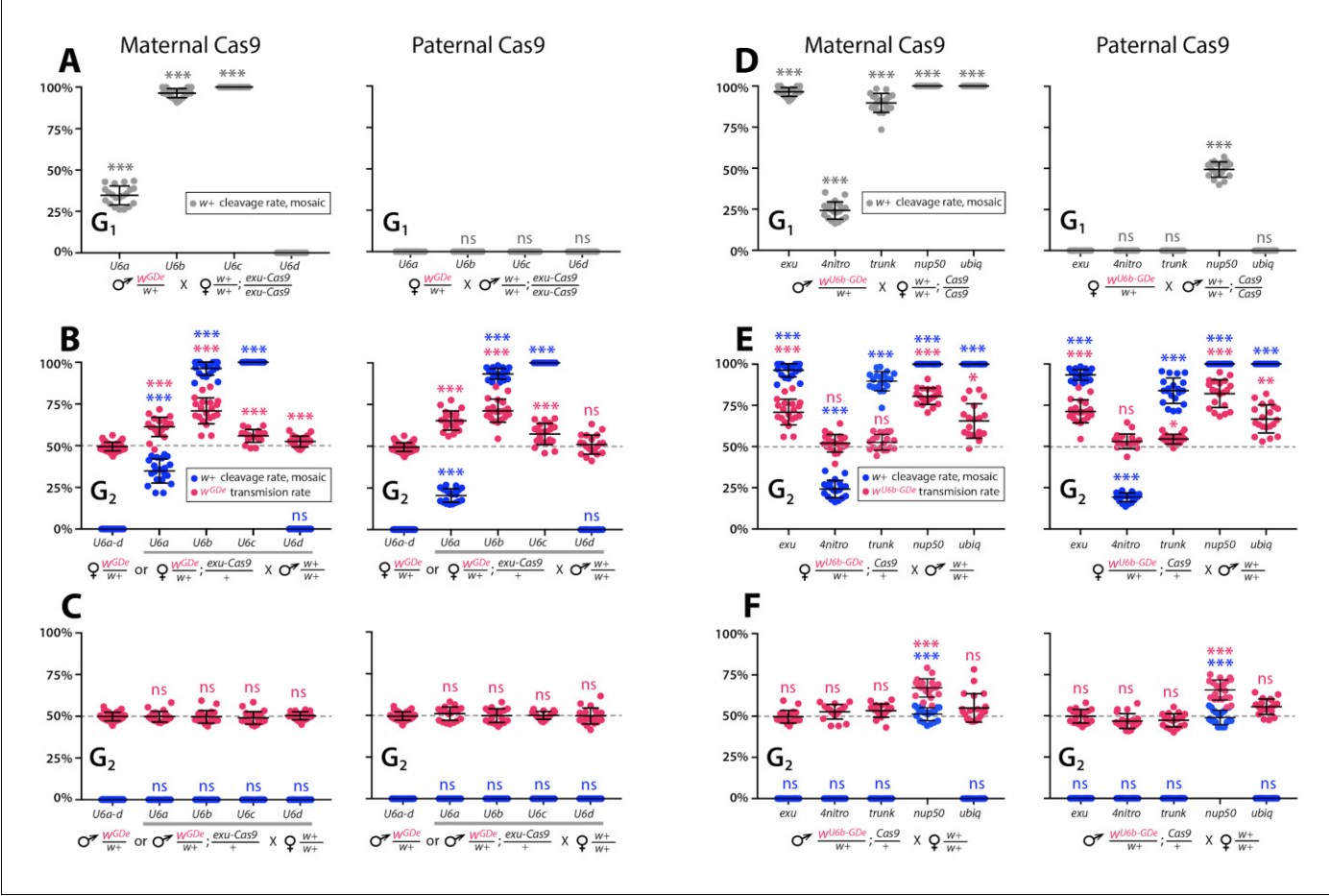

**Figure 3.** Timing and expression of drive components affect cleavage and transmission rates. We bidirectionally crossed trans-heterozygous mosquitoes that contained both components of a gene drive: the gene drive element (GDe) and the Cas9 transgene ($w^{GDe}/w+$; $Cas9/+$). (**Figure 2A**). (A–C) Four $w^{GDe}/w+$ lines, each with a different $gRNA^w$ U6 promoter, were crossed to the $exu$-Cas9 strain to compare cleavage and homing activity. (**A**) Maternally deposited Cas9 protein induced *white* cleavage in $G_1$ trans-heterozygotes harboring $w^{U6a-GDe}$, $w^{U6b-GDe}$, and $w^{U6c-GDe}$ but not $w^{U6d-GDe}$. (**B**) In comparison to $w^{GDe}/w+$, trans-heterozygous $w^{GDe}/w+$; $exu$-Cas9/+ females, (**C**) but not males, exhibited super-Mendelian transmission of $w^{GDe}$. The $w^{U6b-GDe}/w+$; $exu$-Cas9/+ females transmitted $w^{U6b-GDe}$ to 70.9 ± 7.8% of $G_2$ progeny. (D–F) Five lines expressing Cas9 from different promoters were crossed to $w^{U6b-GDe}/w+$. (**D**) Maternally deposited Cas9 resulted in *white* cleavage in $G_1$ trans-heterozygotes. (**E**) Three out of five tested Cas9 lines, $exu$-Cas9, $nup50$-Cas9, and $ubiq$-Cas9, induced super-Mendelian transmission of $w^{GDe}$ by trans-heterozygous females. The $w^{U6b-GDe}/w+$; $nup50$-Cas9/+ females transmitted $w^{U6b-GDe}$ to 80.5 ± 5.0% of $G_2$ progeny. (**F**) All trans-heterozygous males transmitted $w^{GDe}$ following a regular Mendelian inheritance except for the $w^{U6b-GDe}/w+$; $nup50$-Cas9/+ males that induced *white* cleavage in 51.0 ± 3.9% and transmitted the $w^{U6b-GDe}$ allele to 66.9 ± 5.4% of $G_2$ progeny. Point plots show the average ± standard deviation (SD) over 20 data points. Grey dotted line indicates standard Mendelian inheritance rates. Statistical significance was estimated using an equal variance $t$-test. ($P \geq 0.05^{ns}$, $p<0.05^*$, $p<0.01^{**}$, and $p<0.001^{***}$).

The online version of this article includes the following figure supplement(s) for figure 3:

**Figure supplement 1.** Split-drive over multiple generations.
**Figure supplement 2.** Sequences of de novo resistance alleles at *white* locus ($w^R$).

## Cas9-associated variability in homing efficiency

Given the modest homing efficiencies in females, and the lack of homing and cleavage in males developed above, we wanted to further optimize the system to ensure better drive performance. To do this, we varied Cas9 expression and timing to test for improved homing efficiencies. The best performing $w^{U6b-GDe}/+$ line was outcrossed to additional previously developed Cas9 strains (**Li et al., 2017**), each with a unique promoter expressing Cas9: $4nitro$-Cas9 (4-nitrophenyl phosphatase, AAEL007097), $trunk$-Cas9 (trunk, AAEL007584), $nup50$-Cas9 (nucleoporin 50 kDa, AAEL005635), or $ubiq$-Cas9, which enabled us to test the effect of differential Cas9 expression on homing efficiency. Homing and cleavage efficiencies were evaluated in the $G_1$ trans-heterozygotes

$w^{U6b-GDe}/w+$; Cas9/+ with maternally or paternally inherited Cas9 (*Figure 2*). The $G_1$ trans-heterozygotes resulting from outcrosses of these maternal Cas9 lines to U6b-GDe had a somatic mosaic *white* phenotype and cleavage rates of 25.3 ± 6.3% (*4nitro-Cas9*), 85.8 ± 4.6% (*trunk-Cas9*), and 100% (*nup50-Cas9* and *ubiq-Cas9*) (*Figure 3D*, *Supplementary file 4a*). Moreover, zygotic Cas9 activity induced the somatic mosaic *white* phenotypes 49.0 ± 7.1% of the time in $G_1$ trans-heterozygotes from paternally inherited *nup50-Cas9* crosses (*Figure 3D*, *Supplementary file 4b*). Interestingly, homing efficiencies varied by Cas9 strain; for example, trans-heterozygous $w^{U6b-GDe}/w+$; *nup50-Cas9/+* females induced this somatic mosaicism in 100 ± 0% of their $G_2$ progeny with an average 80.5 ± 5.0% inheritance rate in $G_2$ progeny, while 51.0 ± 3.9% of $G_2$ progeny from $w^{U6b-GDe}/w+$; *nup50-Cas9/+* male crosses had this somatic mosaicism and 66.9 ± 5.4% of the $G_2$ progeny inherited the drive (*Figure 3E–F*; *Supplementary files 4c- 4d*). The role of maternal effects in differential drive transmission was assessed by repeating these drive experiments with paternal Cas9 lines in $G_2$, however we did not detect significant differences in drive inheritance between maternal and paternal Cas9 lines (*t*-test with equal variance, p>0.05; *Figure 3E–F*; *Supplementary files 4e- 4f*).

## Multi-generation split drive stability

Gene drives need to have a predictable behavior and stability across many generations, so these properties of the split-drive were determined over multiple consecutive generations by bidirectional outcrosses of $w^{U6b-GDe}/w+$; *exu-Cas9/+* and $w^{U6b-GDe}/w+$; *nup50-Cas9/+* females and males to *wt* lines, which should identify generational variations in cleavage or homing efficiencies indicating instability. In general, the cleavage and transmission rates were relatively stable over multiple generations (*Figure 3—figure supplement 1*). However, super-Mendelian transmission of $w^{U6b-GDe}$ varied between individual mosquito crosses in both split drive systems. For example, the average transmission rate for $w^{U6b-GDe}/w+$; *exu-Cas9/+* females was 68.9 ± 8.9% (range 50–88%) and for $w^{U6b-GDe}/w+$; *nup50-Cas9/+* females was 80.8 ± 8.9% (range 59–94%); (*Figure 3—figure supplement 1*; *Supplementary files 5a- 5b*). Moreover, $w^{U6b-GDe}/w+$; *exu-Cas9/+* males did not induce somatic *white* cleavage nor bias $w^{U6b-GDe}$ transmission (50.0 ± 4.2%; *Figure 3—figure supplement 1A*, *Supplementary file 5c*), while $w^{U6b-GDe}/w+$; *nup50-Cas9/+* males did induce *white* cleavage in 48.5 ± 4.0% of progeny and biased transmission of the $w^{U6b-GDe}$ allele to 67.3 ± 5.2% of progeny over multiple generations (*Figure 3—figure supplement 1B*; *Supplementary file 5d*).

## Discovery of drive-resistant alleles

While tracking drive stability over successive generations, we consistently discovered drive-resistant alleles, which could limit the duration of spread of these drives in real-world settings. For example, $w^{U6b-GDe}/w+$; *nup50-Cas9/+* females transmitted *white* mutant phenotypes to 0.31 ± 0.07% of progeny that did not inherit both split drive transgenes, indicating that drive-resistant mutations were indeed generated at low frequency (*Supplementary file 5b*). To determine whether these mutations occurred in the germline, we outcrossed these individuals to *white* recessive mutants, expecting to see white phenotypes if even one copy of *white* was disrupted in the germline. These experiments yielded 100% white-eye progeny, indicating that trans-heterozygous $w^{U6b-GDe}/w+$; *nup50-Cas9/+* females deposited the Cas9 protein loaded with gRNA$^w$ (Cas9/gRNA$^w$) and this complex induced *white* germline mutations in progeny that did not inherit the $w^{U6b-GDe}$ transgene (*Supplementary file 6*). To confirm this on a molecular level, we performed genomic PCR/sequencing of the *white* locus from individual mutant mosquitoes and found variation in cleavage repair events (*Figure 3—figure supplement 2*). For example, we identified loss-of-function resistance alleles ($w^R$) that frequently contained indels at the gRNA$^w$ target site, making them unrecognizable to subsequent cleavage. This maternal carryover effect of the Cas9/gRNA$^w$ complex was previously described in other drive systems and will likely be an issue when maternal promoters are utilized for drives (*Gantz et al., 2015*; *Kandul et al., 2019b*; *Lin and Potter, 2016*; *Oberhofer et al., 2018*).

## Fitness of gene drive lines

The fitness cost of gene drives and their associated cargo directly impact the behavior and stability of gene drives in wild populations (*Burt, 2003*). Gene drives with high fitness costs are more likely to go to extinction in a population despite strongly biased gene drive inheritance. To evaluate the potential fitness costs associated with the gene drive components, fecundity, egg hatch rate, larval

development time, male competitiveness and adult survival were used as overall indicators of the fitness of the $w^{U6b-GDe}$, nup50-Cas9 and $w^{U6b-GDe}$/nup50-Cas9 lines. There were no significant differences in any of these fitness parameters between the transgenic and wt lines, with the exception of female fecundity (**Supplementary file 7a**). Female fecundity was significantly lower in the nup50-Cas9 and the $w^{U6b-GDe}$/nup50-Cas9 lines compared to the wt lines. Cas9 expression is toxic to many organisms (**Cobb et al., 2015**; **Jiang et al., 2014**; **Wendt et al., 2016**), particularly when it is expressed at high levels (**Cho et al., 2018**), so it is not unexpected that germline expression of Cas9 in the nup50-Cas9 and $w^{U6b-GDe}$/nup50-Cas9 lines may result in the reduced fecundity of these lines. This result is further supported by the fact that there was no significant difference between the $w^{U6b-GDe}$ gRNA only and the wt lines. Reduced fecundity was also only demonstrated in females expressing Cas9, while males were unaffected. Maternal deposition of Cas9 has been shown to be high enough in *Drosophila melanogaster* to facilitate gRNA directed mutagenesis in offspring in the absence of Cas9 inheritance (**Kandul et al., 2019b**; **Lin and Potter, 2016**), so presumably Cas9 can be highly abundant early in embryogenesis and perhaps abundant enough to cause toxicity and reduced fecundity.

## Multi-release of split drives results in high-frequency spread

Our next goal was to evaluate weekly releases of males homozygous for the split drive to determine whether this system could effectively drive an anti-pathogen gene (linked to the gRNA allele) into a population at a high frequency ($\geq$80%) for an extended duration sufficient for disease control. To do this, we used our molecular work to inform the parameters (**Supplementary file 7b**) of simulated split drive releases of the best-performing system described above ($w^{U6b-GDe}$/w+; nup50-Cas9/+). Modeling results suggested that 10 consecutive weekly releases of 10,000 homozygous split drive males into a population with an equilibrium size of 10,000 adults would be sufficient to drive the split drive to a high frequency in the population (~97% having at least one copy of the gRNA/anti-pathogen allele) (**Figure 4A**). Mosquitoes with the gRNA/anti-pathogen alleles would then be slowly eliminated, falling to a frequency of ~87% within three years of the final release. The inheritance bias induced when the Cas9 and gRNA co-occur would give the split drive system an advantage over a comparable inundative release of mosquitoes with the anti-pathogen allele (**Figure 4B**). A hypothetical construct with linked Cas9 and gRNA components and identical homing and fitness cost parameters demonstrated substantial spread after a single release of 10,000 adult males homozygous for the construct but was outnumbered by in-frame resistant alleles four years following the release (**Figure 4C**). The number of in-frame resistant alleles generated could be reduced by carrying out linked system releases of a similar magnitude to those for the split drive system, or by using a linked system with a higher accurate homing rate, perhaps achievable by restricting Cas9 expression to the germline (**Hammond et al., 2018**; **Kandul et al., 2019a**).

## Split drives are persistent and confinable

A key strength of a split drive system is that it is expected to be reversible and largely confinable to a partially isolated population, making it of particular interest for field trials. As a comparison, the hypothetical linked homing drive discussed above, assuming a standard migrant exchange rate of 1% per mosquito per year, spreads to an over 67% allele frequency in the neighboring population following a single release (**Figure 4C**, **Figure 4—figure supplement 1C**). In contrast, the split drive cargo (gRNA/anti-pathogen allele) reaches a peak frequency of ~30% in the neighboring population four years after 10 releases (**Figure 4—figure supplement 1A**) before being gradually eliminated by virtue of the fitness cost. In fact, this fitness cost causes the Cas9 allele to be eliminated in both populations, falling to a frequency of ~10% in the release population within four years post release, and barely reaches a frequency of 2.4% in the neighboring population, followed by a progressive decline of the gRNA/anti-pathogen allele in both populations. The reversal of the split drive system can be accelerated by the release of wild-type males. After just 10 releases, the frequency of the split drive system in the population is expected to be sufficient for interrupting local disease transmission over a three year period. **Figure 4D** depicts the duration for which the anti-pathogen allele is predicted to remain in over 80% of female mosquitoes when the number of releases and fitness cost of the gRNA/anti-pathogen allele are varied. This duration exceeds three years for the release scheme described above, and for fitness costs less than 10% in gRNA/disease-refractory allele homozygotes.

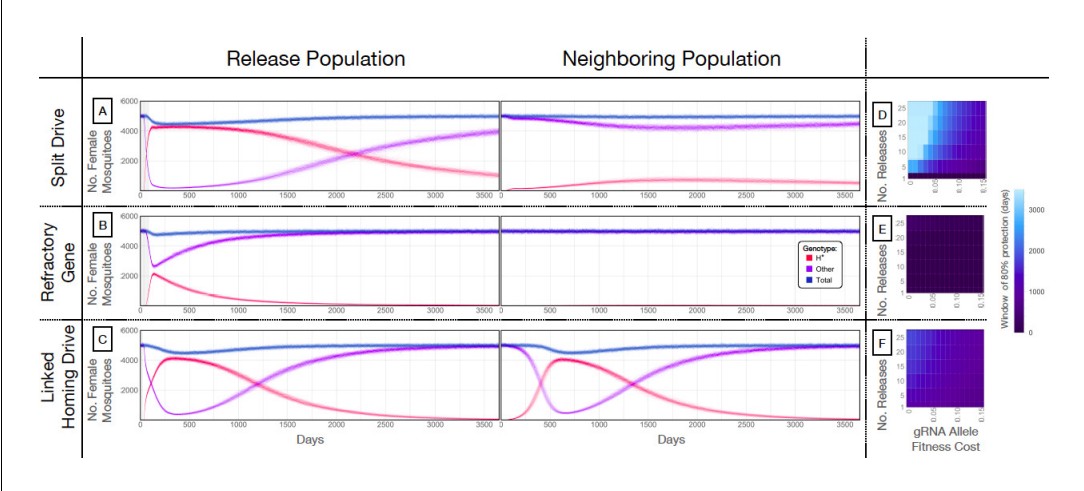

**Figure 4.** Mathematical model predictions for best performing split drive. Model predictions for releases of *Ae. aegypti* mosquitoes homozygous for (A) the split drive system, (B) a disease-refractory gene, or (C) a homing drive system in which the components of the split drive system are linked at the same locus. Parameters correspond to those for the best performing split drive system ($w^{U6b-GDe}/w+$; *nup50-Cas9/+*) (**Supplementary file 7b**). Releases are carried out in a population with an equilibrium size of 10,000 adults and a 1% per mosquito per generation migration rate with a neighboring population of the same equilibrium size. Model predictions were computed using 100 realizations of the stochastic implementation of the MGDrivE simulation framework (*Sánchez et al., 2018*). Weekly releases of 10,000 homozygous split drive males or the disease-refractory gene were simulated over a 10 week period, while a single release was simulated for the linked homing drive system. Total female population size ('total', dark blue), adult females with at least one copy of the disease-refractory allele ('H*', red), and disease-susceptible adult females without the disease-refractory allele ('other', purple) were plotted for each group. Notably, the split drive system is: i) largely confined to its release population, ii) reversible, and iii) present at a high frequency (>85% of adult females having at least one copy) for over three years. The split drive system outperforms inundative adult male release of the disease-refractory gene in population disease refractoriness and outperforms the confinability of a linked homing drive system. In the right column, heatmaps are shown for (D) the split drive system, (E) inundative releases of a disease-refractory gene, and (F) a linked homing drive system, and depict the window of protection in days that the proportion of mosquitoes in the release population with at least one copy of the disease-refractory allele exceeds 80%. The fitness cost (reduction in mean adult lifespan) associated with gRNA/refractory allele homozygotes is varied along the x-axis, and the number of weekly releases along the y-axis. Notably, for the split drive system, the window of protection exceeds three years following 10 or more weekly releases for gRNA/refractory allele fitness costs of 10% in homozygotes.

The online version of this article includes the following figure supplement(s) for figure 4:

**Figure supplement 1.** Allele frequencies for best performing split-drive construct.

**Figure supplement 2.** Fecundity of trans-heterozygous females only slightly reduced in mathematical models.

## Discussion

*Ae. aegypti* is a globally distributed vector that spreads deadly pathogens to millions of people annually. Current control methods are inadequate for this vector and therefore new innovative technologies need to be developed and implemented. With the aim of providing new tools for *Ae. aegypti* control, here we engineer and test the first gene drives developed in this species. Importantly, to make these drives safe and confineable, we engineered these systems as split drives. These drives functioned at very high efficiency and may provide a tool to control this vector. To develop these drives, initially we functionally characterized several drive components that performed with high efficiency in both somatic cells and the germline. Additionally, we discovered a drive target sequence, within the *white* gene, that is highly conserved across many geographically distinct populations, providing a useful phenotypic readout of drive functionality, and demonstrating that highly conserved drive target sites can be found through extensive genomic sequencing of wild populations. Multiple bidirectional genetic crosses of transgenic lines with optimized drive components revealed that homing and cleavage efficiencies are largely dependent on the timing and expression of drive components and maternal deposition can result in embryonic activity and the generation of drive resistant alleles. Despite the creation of resistant alleles, however, mathematical modeling predicts that these split drives can spread locally and are also confinable and reversible, making them suitable for field trials and capable of local disease control once effectors are linked.

Notably, components characterized in this study including the polymerase III U6 promoters crossed with robustly Cas9 expression strains, can also be used to accelerate basic functional gene analysis in *Ae aegypti*. For example, tissue specific Cas9 strains could be crossed to U6-driven gRNA strains disrupting essential genes, to enable biallelic mutations in specific tissues (e.g. neurons) which otherwise would not be possible due to lethality if disrupted in all cells. Additionally, if linked, these components could generate a potent drive that could self-disseminate for wider regional mosquito control. Our models predict that if these components were directly linked, assuming homing and cleavage rates similar to when unlinked, the drive would provide a window of >80% disease resistance among the female *Ae. aegypti* population for several years following a release (*Figure 4F*) and spread from one population to another (*Figure 4C*). These components could also be used in the future to develop self-limiting population suppression technologies, such as a precision-guided sterile insect technique (*Kandul et al., 2019b*), or to target conserved sex-determination genes such as doublesex (*dsx*) to generate a potent homing-based population suppression drive similar to the drive recently developed for *Anopheles gambiae* (*Kyrou et al., 2018*). Alternatively, these components could be used to generate a novel population suppression design by encoding a functional copy of *Nix* (*Aryan et al., 2019*; *Hall et al., 2015*), as the cargo, thereby converting females into fertile males while the drive spreads through and suppresses the population. Finally, these components support the development of other drives, including *trans*-acting toxin- and *cis*-acting antidote- (TA) based drives (*Kandul et al., 2019b*; *Oberhofer et al., 2019*), evolutionarily stable homing-based replacement drives that target haplosufficient essential genes and encode a recoded (i.e., cleavage resistant) rescue of the target gene (*Champer et al., 2016*; *Kandul et al., 2019b*) or even TA underdominance-based drives (*Champer et al., 2016*), all of which could be instrumental for spreading anti-pathogen effectors (e.g., *Buchman et al., 2019*) into wild disease-transmitting populations.

While this is a major step towards developing field-worthy gene-drive systems in *Ae. aegypti*, there are still limited effectors available to link with population modifying drives that will limit the immediate use of these technologies. Most antiviral effectors developed in *Ae. aegypti* to date rely upon small RNA pathway-directed targeting of viral genomes (*Buchman et al., 2019*; *Franz et al., 2014*; *Franz et al., 2006*; *Mathur et al., 2010*; *Yen et al., 2018*); however, these antiviral transgenes are not easily multiplexed and the diversity of arboviruses transmitted by this vector as well as the high mutation rate of these viruses (*Steinhauer et al., 1992*) makes their practicality in the field tenuous. New, innovative effectors are in development, for example, the recent broad neutralizing antibody-based anti-dengue effector (*Buchman et al., 2019*), but long-term studies of viral resistance to all current antiviral effectors are limited, so their long-term effectiveness is unknown *Marshall et al., 2019*). Furthermore, effector-associated fitness costs could impact the success of these technologies. While drive technologies should overcome some of these fitness costs, as we balance the optimization of drive efficacy with the minimization of drive risks, we will likely still need to develop effectors with reduced fitness costs. Current effectors have achieved efficacy against only a limited number of viral families and strains, usually those which are most highly lab adapted, and may have concerning fitness costs (Marshall et al. 2019). Therefore, additional resources are needed to develop antiviral effectors that have reduced fitness costs, achieve broad antiviral activity against the diversity of viruses vectored by *Ae. aegypti,* and limit viral resistance despite high viral mutation rates.

It should be pointed out that gene-drive technologies are imperfect. Drive efficiencies are impacted by target-site availability, timing, the expression of drive components, host-specific factors such as the timing of meiosis and recombination, as well as the generation of resistant alleles that form as a result of non-homologous end-joining (NHEJ) repair of CRISPR-mediated DNA cleavage as opposed to homology directed repair (HDR). Notwithstanding, significant efforts are underway to optimize next generation drives that incorporate molecular design architectures predicted to mitigate the generation and selection of resistant alleles and improve conversion efficiencies. These efforts include rational target-site design, regulation of the expression and timing of the nuclease, multiplexing drive targets, improved regulatory elements, and targeting ultra-conserved regions in essential genes that are recorded in the drive, enabling the selective elimination of non-functional NHEJ-repaired alleles (*Hammond and Galizi, 2017*; *Raban and Akbari, 2017*; *Champer et al., 2016*; *Marshall et al., 2017*; *Kandul et al., 2019a*; *Kandul et al., 2019b*; *Oberhofer et al., 2019*). While these future drive designs are expected to be more efficient than current systems, it should be noted that even imperfect drives with modest homing efficiencies are still predicted to be quite

invasive (*Noble et al., 2017*), and will likely be useful for field implementation which could confer long-lasting entomological and epidemiological impacts.

Given the remarkable progress in the gene drive field and potential of gene drives to spread globally, understandably the discussion of the ethics, risks, governance, and the guidance of gene drives in the field have come to the forefront (*Adelman et al., 2017*; *Akbari et al., 2015a*; *James et al., 2018*; *Kaebnick et al., 2016*). As gene-drive technologies advance and field trials become a necessary step toward understanding the efficacy and behavior of drives in wild populations, it is important that facilities planning and management, infrastructure upgrades, talent development for deployment and monitoring, and public engagement are advanced to support the sustainability of gene drive technologies. Safe, non-invasive, self-limiting split drive technologies, such as the systems developed here, are a responsible choice for studying gene drives in the wild, as they safeguard against spreading to non-target populations and will allow the assessment of potential risks and unintended consequences. This may help establish the necessary infrastructure, while simultaneously controlling disease burdens, in preparation for a potential subsequent release of an efficient linked-drive system that can self-disseminate catalytically into broader landscapes. Taken together, our results provide compelling evidence for the feasibility of future effector-linked split drive technologies that can contribute to the safe, sustained control and potentially the elimination of pathogens transmitted by this species.

# Materials and methods

## Insect rearing

Mosquitoes used in all experiments were derived from the *Ae. aegypti* Liverpool strain (designated as wild-type [*wt*], *Supplementary file 8a*). Of note, the *Ae. aegypti* reference genome sequences were generated using this strain (*Matthews et al., 2018*; *Nene et al., 2007*). Mosquitoes were raised in incubators at 28.0°C with 70–80% humidity and a 12 hr light/dark cycle. Larvae were fed ground fish food (TetraMin Tropical Flakes, Tetra Werke, Melle, Germany) and adults were provided 0.3 M aqueous sucrose ad libitum. Adult females were blood fed three to five days after eclosion using anesthetized mice. Mosquitoes were examined, scored, and imaged using the Leica M165FC fluorescent stereo microscope equipped with the Leica DMC2900 camera. All animals were handled in accordance with the Guide for the Care and Use of Laboratory Animals as recommended by the National Institutes of Health and supervised by the local Institutional Animal Care and Use Committee (S17187).

## Profile of natural variation at *white* target site

To assess the natural variation within the *white* target site, we used data from individual whole genome sequencing of 143 *Ae. aegypti* specimens sampled in California (N = 127), Florida (N = 4), Puerto Rico (N = 6), Mexico (N = 3), and South Africa (N = 3) (*Figure 1—figure supplement 1*, *Figure 1—source data 1*) and sequenced to an approximate depth of 10x on an Illumina HiSeq 4000 producing 17.1 giga base pairs (Gbp). Then 17.1 Gbp of sequence reads were mapped with BWA-MEM v0.7.15 (*Li, 2013*) to the latest version of the reference genome of *Ae. aegypti* (AaegL5 [*Matthews et al., 2018*]). Polymorphisms were called with Freebayes v1.0.1 (*Garrison and Marth, 2012*), applying all the default parameters except for 'theta = 0.01' and 'max-complex-gap=3'. We employed the most conservative approach by applying no filtering to the called polymorphisms to maintain the broadest possible set of potential variation. Additionally, we checked the polymorphism data for *Ae. aegypti* available on https://www.vectorbase.org (accessed 1/2/19) and found no published SNP's in the region of the *white* target site. We therefore consider the target site likely conserved in natural populations.

## Construct design and assembly

The Gibson enzymatic assembly method was used to build all constructs. To generate the gRNA constructs, the *piggyBac* plasmid (*Akbari et al., 2015b*) was digested with FseI and AscI, and the linearized fragment was used as a backbone for construct assembly. Constructs for the U6 promoter screen contained the following fragments: the 5'-end flanking sequence of the U6 snRNA, which was selected as the promoter region; a 20-base sequence of the *gRNA^w* (*Figure 1—figure supplement*

*2A*); a 76-base fragment with a gRNA scaffold; and a 3xP3-eGFP-SV40 fragment. The first three fragments were generated using commercial gene synthesis (gBLOCK by IDT and GenScript). The 3xP3-eGFP fragment was amplified from a previously described plasmid harboring the 3xP3 promoter and coding sequence of eGFP (Addgene #104968 and 100705) using primers AE01, AE02, AE03, and AE04 (*Supplementary file 8a*, *Supplementary file 8b*). A total of six plasmids were generated and are referred to as follows: U6a-T, U6b-T, U6c-T, U6d-T, U6e-T andU6f-T (*Figure 1—figure supplement 2C*, *Supplementary file 8a*).

We engineered four gene-drive element (GDe) constructs carrying different U6 promoters for site-specific integration at the *white* locus: *U6a-GDe*, *U6b-GDe*, *U6c-GDe*, and *U6d-GDe*. Each plasmid contained the following fragments (*Figure 1—figure supplement 2D*): (1) left and right homology arms of ~1 kb in length, which are complementary to the *Ae. aegypti white* locus immediately adjacent to the 5' and 3' ends of the $gRNA^w$ cut site, respectively, were gene synthesized by GenScript; (2) a U6 promoter with a 20-base $gRNA^w$ sequence and a 76-base gRNA scaffold, which was PCR amplified from U6a-T, U6b-T, U6c-T, and U6d-T constructs with the following primers: AE05 and AE06, AE07 and AE08, AE09 and AE10, AE11 and AE12, respectively; (3) the 3'UTR region of the corresponding U6 snRNA was gene synthesized by GenScript; (4) a 3xP3-tdTomato-SV40 fragment, which was amplified from the previously described plasmid (Addgene #104968) using the following primers: AE13 and AE14, AE15 and AE14, AE16 and AE14, AE17 and AE14; (5) an Opie2-dsRed-SV40 fragment, which was amplified from the previously described plasmid (Addgene #100705) using primers AE18 and AE19 (*Supplementary file 8a*, *Supplementary file 8b*). All plasmids were grown in JM109 chemically competent cells (Zymo Research #T3005) and isolated using Zyppy Plasmid Miniprep (Zymo Research #D4037) and Maxiprep (Zymo Research #D4028) kits (*Supplementary file 8a*). Each construct sequence was verified using Source Bioscience Sanger sequencing services. A list of primer sequences used in the above construct assembly can be found in *Supplementary file 8b*. We have also made all plasmids and sequence maps available for download and/or order at Addgene (www.addgene.com) with identification numbers listed in *Figure 1—figure supplement 2*.

## Embryo microinjection and mutation screening

Embryonic collection and CRISPR microinjections were performed following previously established procedures (*Aryan et al., 2014*; *Kistler et al., 2015*). The concentration of plasmids used for the U6 promoter screen was 300 ng/μL. Injected $G_0$ and $G_1$ progeny were visualized at the larval, pupal, and adult life stages under a dissecting microscope (Olympus SZ51 and Leica M165FC). The heritable mutation rates were calculated as the number of $G_1$ progeny with the loss-of-function mutation out of the number of all $G_1$ progeny crossed with the white eye (*w–*) strain mosquitoes. To integrate each GDe construct at the *white* locus, a mixture containing 100 ng/μL of synthetic $gRNA^w$, 100 ng/μL of each U6-GDe plasmid (U6a-GDe, U6b-GDe, U6c-GDe, and U6d-GDe), and 100 ng/μL of Cas9 protein was injected into 500 *wt* embryos for each plasmid. Synthetic gRNAs (Synthego) and recombinant *Streptococcus pyogenes* Cas9 protein (PNA Bio Inc, *Supplementary file 8a*) were obtained commercially and diluted to 1,000 ng/μL in nuclease-free water and stored in aliquots at –80℃. A total of 233, 271, 191, 215 $G_0$ adults were recovered for U6a-, U6b-, U6c-, and U6d-GDe injections, respectively. Successful integration into the *white* locus was determined by visually identifying the eye-specific 3xP3-tdTomato fluorescence in $G_1$ heterozygous mosquito larvae with black eyes ($w^{U6-GDe}/w+$) and in $G_2$ homozygous mosquito larvae with white eyes ($w^{U6-GDe}/w^{U6-GDe}$). In addition, site-specific integration of U6-GDe constructs was confirmed by amplifying and Sanger sequencing both the left and right integration points (*Figure 2—figure supplement 2*) from a genomic DNA prep of each $w^{U6-GDe}$ line with the following primers: AE20, AE21, AE22, and AE23 (*Supplementary file 8b*).

## Assessment of split gene-drive efficacy

Both gene-drive (*GD*) and *Cas9* elements are required for a functional split drive. To generate $G_1$ trans-heterozygous $w^{U6-GDe}/w+$; *Cas9/+* male and female mosquitoes with an active split drive, we reciprocally crossed $w^{U6-GDe}/w+$ and *Cas9/Cas9* strains. While selecting trans-heterozygous mosquitoes, we scored the loss-of-function mutation of *white* in all $G_1$ progeny. Then, $G_1$ trans-heterozygous mosquitoes were mated to *wt* for 4 days, and the *white* mutation and $w^{U6-GDe}$ transmission frequencies were calculated in $G_2$ progeny. Females were blood fed on anesthetized mice on the

fifth day. Three days later, more than twenty females were allowed to lay eggs individually into a 50 mL vial filled with water and lined with wet filter paper. The larval progeny from each female were counted and scored for $w^{U6-GDe}$ and Cas9 by 3xP3-tdTomato and Opie2-dsRed expression, respectively. Females that failed to take blood and produce progeny were excluded from the analysis. According to Mendelian inheritance, the expected $G_2$ transmission rate of $w^{U6-GDe}$ is 50% under this crossing scheme; the higher rate indicates super-Mendelian inheritance (Figure 2A). White mutagenesis and $w^{U6-GDe}$ transmission rates by trans-heterozygous $w^{U6-GDe}/w+$; Cas9/+ mosquitoes each carrying GDe with four different U6 promoters were compared to those of the corresponding $w^{U6-GDe}/w+$ line without Cas9 in the genetic background. The homing and cleavage efficiencies of the two split-drive systems, $w^{U6b-GDe}/w+$; exu-Cas9/+ and $w^{U6b-GDe}/w+$; nup50-Cas9/+, were observed for the following four generations by scoring the markers in the progeny from reciprocal crosses of trans-heterozygous and wt mosquitoes.

## Molecular characterization of $w^R$ alleles

To test whether $w^R$-resistant alleles were present in germ cells, mosquitoes harboring $w^R$ alleles were genetically crossed into the $w^{19\Delta}/w^{19\Delta}$ recessive mutant background, and their progeny was scored for white loss-of-function mutations. Genomic DNA was extracted from an individual mosquito of each progeny group using the DNeasy Blood and Tissue Kit (QIAGEN) following the manufacturer's protocol, and a white region carrying the $gRNA^w$ target site was PCR amplified with primers AE24 and AE25. Because the tested mosquitoes were heterozygous $w^R/w^{19\Delta}$ for white alleles, a PCR amplicon from each mosquito was first cloned into an open plasmid using Gibson enzymatic assembly, and seven clones were PCR amplified and Sanger sequenced to identify novel $w^R$ alleles. De novo-mutated $w^R$ alleles were identified by comparing them with both $w+$ and $w^{19\Delta}$ alleles using Sequencher and 5.0 SnapGene 4.2. The same method was used to confirm the presence of $w+$ non-cleaved alleles in a few mosquitoes with black eyes, which were later inferred to be $w+/w^{19\Delta}$ and $w+/w^{U6b-GDe}$.

## Fitness of transgenic strains

To determine if the presence of a transgene(s) conferred a fitness cost, multiple fitness parameters were evaluated between $w^{U6b-GDe}/nup50-Cas9$, $w^{U6b-GDe}$; nup50-Cas9 and wt mosquitoes. Insectary conditions were as described in the insect rearing section. Fecundity and fertility were assessed from reciprocal male and female crosses to the wt strain. For the female fecundity and fertility assessments, 20 females were mated to 20 wt males and for the male fecundity and fertility assessments, single males were mated to 20 wt females. After four days, female mosquitoes were blood fed until fully engorged and individually transferred into plastic vials with tap water and lined with an egg paper. Mosquitoes were allowed to oviposit for three days and after fecundity was assessed as the number of eggs laid per female. Eggs were stored in the insectary for four days to allow full embryonic development then hatched in a vacuum chamber. Larvae were counted on the third day post hatch. Fertility was calculated as the percentage of eggs hatched per female. An analysis of variance (ANOVA) and Tukey post-hoc test were performed to compare differences in fecundity and fertility between all groups. To assess larval to pupal development time, transgenic and wt were distributed into pans, at a density of 10 larvae per pan, containing 2.5 L of ddH2O and 0.6 mL of 1:1 Tetramin fish food:water. Larvae were counted twice daily until pupation and then date of pupation and emergence were recorded. Larval development time was calculated for each sex as there are sex-specific differences in larval development time in this species. An ANOVA and Tukey post-hoc test were performed to compare differences in larval development between all groups. Male mating competitiveness was assessed by 1:1 transgenic male:wt male competition studies in 30 small cages. Transgenic and wt males 2 days post emergence and raised under the same standardized larval conditions were placed into a small cage containing a female wt mosquito. Females were allowed to acclimate to the cage for 72 hours prior to male release. Mating, blood feeding and oviposition, as well as egg collection and hatching conditions, were as described for the fertility and fecundity studies. Larvae at three days post hatch were screened for transgene markers under a Leica M165FC fluorescent microscope. Females with transgenic offspring were scored as mated to transgenic males and females with no transgenic offspring were scored as mated to wt males. All females produced eggs. A chi-squared test was used to determine whether there were differences in the number of observed

and expected matings to a transgenic male. To assess adult mosquito survival, one cage of each transgenic or *wt* strain containing equal numbers of male and females were monitored daily for survival until all mosquitoes had died. Median time to death and differences between transgenic and *wt* survival curves were calculated by a Mantel-Cox log rank test. Due to sex specific differences in survival, analyses also were separated by sex. All statistical analyses were performed using GraphPad Prism software (GraphPad Software, La Jolla, California, USA). P values > 0.05 were considered not significant.

## Mathematical modeling

To model the expected performance of a split drive functioning as a confinable and reversible gene-drive system and as a test system prior to the release of a linked homing drive for *Ae. aegypti*, we simulated release schemes for split drive, linked homing drive, and inundative releases of males carrying a refractory allele using the MGDrivE simulation framework (*Sánchez et al., 2018*) (https://marshalllab.github.io/MGDrivE/). This framework models the egg, larval, pupal, and adult mosquito life stages (both male and female adults are modeled) implementing a daily time step, overlapping generations, and a mating structure in which adult males mate throughout their lifetime while adult females mate once upon emergence, retaining the genetic material of the adult male with whom they mate for the duration of their adult lifespan. Density-independent mortality rates for the juvenile life stages are assumed to be identical and are chosen for consistency with the population growth rate in the absence of density-dependent mortality. Additional density-dependent mortality occurs at the larval stage, the form of which is taken from previous studies (*Deredec et al., 2011*). The inheritance patterns for the split drive, linked homing drive, and refractory gene systems are modeled within the inheritance module of the MGDrivE framework (*Sánchez et al., 2018*) along with their impacts on female fecundity and adult mortality rate. We parameterized our split drive model according to the best-performing system in this study ($w^{U6b-GDe}/w+$; *nup50-Cas9/+*): i) a cleavage frequency of 100.0% in females and 51.0% in males, ii) HDR frequency 80.5% in females and 66.9% in males, and iii) each Cas9 allele was associated with a 7.8% reduction in female fecundity (*Figure 4—figure supplement 2*). Ballpark parameter estimates were used in cases where direct measurements were not available: i) a sixth of non-HDR events were assumed to lead to in-frame/cost-free resistant alleles (i.e. half of the mutations that preserve the reading frame), with the remaining leading to out-of-frame/costly resistant alleles, and ii) by default, we assumed a 10% reduction in mean lifespan associated with gRNA/disease-refractory allele homozygotes, and no fitness cost in heterozygotes, although a range of values were explored for this parameter. We implemented the stochastic version of the MGDrivE framework to capture the randomness associated with rare events such as resistant allele generation. We simulated two partially isolated populations each consisting of 10,000 adults at equilibrium, exchanging migrants at a rate of 1% per mosquito per generation, with weekly releases of 10,000 adult males over a defined period. *Ae. aegypti* life history and intervention parameter values are listed in *Supplementary file 7b*.

## Statistical analysis

Unless described previously, statistical analysis was performed in JMP 8.0.2 by SAS Institute Inc. Twenty replicates were used to generate statistical means for comparisons. *P* values were calculated for a two-sample Student's t-test with equal variance. Point plots were built in Prism 7 by GraphPad.

## Acknowledgements

This work was supported by funding from a Defense Advanced Research Project Agency (DARPA) Safe Genes Program Grant (HR0011-17-2- 0047) awarded to OSA and subcontracted to GCL and JMM. We acknowledge funding support from the UC Davis Bridge Funding Program, UC Davis School of Veterinary Medicine Vector-Borne Disease Pilot Grant Program, and the Pacific Southwest Regional Center of Excellence for Vector-Borne Diseases funded by the U.S. Centers for Disease Control and Prevention (Cooperative Agreement 1U01CK000516). We thank Judy Ishikawa for help rearing and maintaining all mosquito strains produced in this study. We also thank Allison Weakley, Kendra Person, and Hans Gripkey for field mosquito collection data processing including DNA extraction, DNA quantification, and library preparations. We thank personnel from Consolidated Mosquito Abatement District (Ms. Jodi Holeman and Ms. Katherine Ramirez), Coachella Valley,

Delta, Greater LA County (Dr. Susan Kluh), and San Mateo County Vector Control Districts and Fresno, Kern, Madera County, Northwest (Dr. Major Dhillon, District Manager), Orange County (Mr. Michael Hearst, District Manager), and San Gabriel Valley Mosquito and Vector Control Districts, Kings Mosquito Abatement District, Community Health Division of the Department of Environmental Health (Ms. Rebecca Lafreniere, Chief and Ms. Elizabeth Pozzebon, Director), Imperial County Public Health Department, San Bernardino County Mosquito and Vector Control Program, San Diego County Dept. of Environmental Health, Vector Control, Dr. Chelsea Smartt (Florida Medical Entomology Laboratory), and Dr. Christopher Barker (UC Davis) for providing specimens used in this study. We also thank Dr. Anthony J Cornel (UC Davis) and Dr. Leo Braack (University of Pretoria, South Africa) for the collection of South African samples and Dr. Danny Governer and SANParks for permitting collection from Shingwedzi in the Kruger National Park. We also thank Dr. Lutz Froenicke and his team at the UC Davis DNA Technologies Core for genome sequencing.

## Additional information

### Funding

| Funder | Grant reference number | Author |
|---|---|---|
| Defense Advanced Research Projects Agency | HR0011-17-2- 0047 | Omar S Akbari |
| Centers for Disease Control and Prevention | 1U01CK000516 | Gregory C Lanzaro |

The funders had no role in study design, data collection and interpretation, or the decision to submit the work for publication.

### Author contributions

Ming Li, Conceptualization, Formal analysis, Investigation, Writing - original draft, Writing - review and editing; Ting Yang, Formal analysis, Investigation; Nikolay P Kandul, Conceptualization, Formal analysis, Writing - original draft, Writing - review and editing; Michelle Bui, Stephanie Gamez, Investigation; Robyn Raban, Formal analysis, Writing - original draft, Project administration, Writing - review and editing; Jared Bennett, Héctor M Sánchez C, Gregory C Lanzaro, Formal analysis; Hanno Schmidt, Formal analysis, Writing - original draft; Yoosook Lee, Formal analysis, Writing - original draft, Writing - review and editing; John M Marshall, Data curation, Formal analysis, Writing - original draft; Omar S Akbari, Conceptualization, Formal analysis, Supervision, Funding acquisition, Writing - original draft, Project administration, Writing - review and editing

### Author ORCIDs

Ming Li (ID) https://orcid.org/0000-0002-7578-4968
Ting Yang (ID) https://orcid.org/0000-0001-7201-4231
Robyn Raban (ID) https://orcid.org/0000-0002-5648-6770
Hanno Schmidt (ID) http://orcid.org/0000-0001-8915-891X
John M Marshall (ID) https://orcid.org/0000-0003-0603-7341
Omar S Akbari (ID) https://orcid.org/0000-0002-6853-9884

### Ethics

Animal experimentation: All animals were handled in accordance with the Guide for the Care and Use of Laboratory Animals as recommended by the National Institutes of Health and supervised by the local Institutional Animal Care and Use Committee (S17187).

### Decision letter and Author response

Decision letter https://doi.org/10.7554/eLife.51701.sa1
Author response https://doi.org/10.7554/eLife.51701.sa2

# Additional files

## Supplementary files

• Supplementary file 1. Frequency of *white* somatic knockout in $G_1$ gRNA[w]/+; Cas9/+ trans-heterozygotes depends on U6 promoter driving gRNA[w] expression.

• Supplementary file 2. Genetic linkage of Gene Drive element (GDe) integrated at *white* locus and *Nix* (male dominant locus), raw data.

• Supplementary file 3. Split drive cleavage and transmission rate. (**a**) Eye phenotypes in $G_1$ progeny from ♂$w^{GDe}$/w+ X ♀ *exu-Cas9/exu-Cas9*. (**b**) Eye phenotypes in $G_1$ progeny from ♀$w^{GDe}$/w+ X ♂*exu-Cas9/exu-Cas9*. (**c**) Eye phenotypes in $G_2$ progeny from trans-heterozygous ♀$w^{GDe}$/w+; *exu-Cas9/+* with Maternal Cas9 X ♂w+/w+ (*wt*). (**d**) Eye phenotypes in $G_2$ progeny from trans-heterozygous ♂$w^{GDe}$/w+; *exu-Cas9/+* with Maternal Cas9 X ♀w+/w+ (*wt*). (**e**) Eye phenotypes in $G_2$ progeny from trans-heterozygous ♀$w^{GDe}$/w+; *exu-Cas9/+* with Paternal Cas9 X ♂w+/w+ (*wt*). (**f**) Eye phenotypes in $G_2$ progeny from trans-heterozygous ♂$w^{GDe}$/w+; *exu-Cas9/+* with Paternal Cas9 X ♀w+/w+ (*wt*).

• Supplementary file 4. Cleavage and homing efficiency variability between Cas9 lines. (**a**) Eye phenotypes in $G_1$ progeny from ♂$w^{U6b-GDe}$/w+ X ♀*Cas9/Cas9* (**b**) Eye phenotypes in $G_1$ progeny from ♀$w^{U6b-GDe}$/w+ X ♂*Cas9/Cas9*. (**c**) Eye phenotypes in $G_2$ progeny from ♀$w^{U6b-GDe}$/w+; *Cas9/+* with Maternal Cas9 X ♂w+/w+ (*wt*). (**d**) Eye phenotypes in $G_2$ progeny from ♂$w^{U6b-GDe}$/w+; *Cas9/+* with Maternal Cas9 X ♀w+/w+ (*wt*). (**e**) Eye phenotypes in $G_2$ progeny from ♀$w^{U6b-GDe}$/w+; *Cas9/+* with Paternal Cas9 X ♂w+/w+ (*wt*). (**f**) Eye phenotypes in $G_2$ progeny from ♂$w^{U6b-GDe}$/w+; *Cas9/+* with Paternal Cas9 X ♀w+/w+ (wt).

• Supplementary file 5. Multi-generation split drive stability. (a) Eye phenotypes in G3, G4, G5 progeny from ♀$w^{U6b-GDe}$/w+; *exu-Cas9/+* X ♂w+/w+ (wt). (b) Eye phenotypes in G3, G4, G5 progeny from ♀$w^{U6b-GDe}$/w+; *nup50-Cas9/+* X ♂w+/w+ (wt). (c) Eye phenotypes in G3, G4, G5 progeny from ♂$w^{U6b-GDe}$/w+; *exu-Cas9/+* X ♀w+/w+ (wt). (d) Eye phenotypes in G3, G4, G5 progeny from ♂$w^{U6b-GDe}$/w+; *nup50-Cas9/+* X ♀w+/w+ (wt).

• Supplementary file 6. Eye phenotypes in progeny from the w- mosquitoes sampled at G2, G3, G4, and G5 (w[U6b-GDe]/w+; nup50-Cas9/+ X w+/w+) and w∆19/w∆19 reference mosquitos.

• Supplementary file 7. Transgene and model fitness parameters. (**a**) Effect of transgenes on fitness. Comparisons of several fitness parameters between wildtype (WT) and transgenic lines expressing gRNAs targeting the w gene (wU6b-GDe) only, Cas9 only (nup50-Cas9) or both gRNAs targeting the w gene and Cas9 (wU6b-GDe/nup50-Cas9). The transgenic and WT mosquitoes only differed significantly in one fitness parameter, female fecundity, which suggests that the transgenic strains do not have a major impact on mosquito fitness. The survivorship of transgenic versus WT mosquitoes is also shown. (**b**) Parameter values used in *Ae. aegypti* population model. (*Carvalho et al., 2015*; *Christophers, 1960*; *Focks et al., 1993*; *Horsfall, 1972*; *Otero et al., 2006*; *Simoy et al., 2015*; *Taylor et al., 2001*).

• Supplementary file 8. Methods supporting information. (**a**) Key resources table (**b**) Primer sequences used in this study.

• Transparent reporting form

## Data availability

All data generated or analysed during this study are included in the manuscript and supporting files.

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
