## [Decision Letter]

**Acceptance summary:**

Li et al. describe the first two-component 'split-drive' system for *Aedes aegypti*. They provide a proof-of-principle example of drive activity. They identify combinations of U6-sgRNA lines and Cas9 expression lines that generate somatic and germline mutagenesis and support super-Mendelian inheritance of the U6-sgRNA drive component. Strains exhibiting these properties might be effective tools for mosquito control while retaining self-limiting properties. This is a significant advance for the field. A caveat is that it is based on a single gene proof of principle. In summary, the paper reports a successful proof-of-principle of a genetically encoded CRISPR/Cas9 gene drive in *Aedes aegypti*.

**Decision letter after peer review:**

Thank you for submitting your article "Development of a confinable gene-drive system in the human disease vector, *Aedes aegypti*" for consideration by *eLife*. Your article has been reviewed by two peer reviewers, and the evaluation has been overseen by a Reviewing Editor and a Senior Editor. The reviewers have opted to remain anonymous.

The reviewers have discussed the reviews with one another and the Reviewing Editor has drafted this decision to help you prepare a revised submission. From an experimental point of view, both reviewers have emphasized the need to measure the fitness cost of these transgenic lines. They argue that this can be done in a timely period.

Summary:

In their manuscript, "Development of a confinable gene-drive system in the human disease vector, *Aedes aegypti*," Li and colleagues describe a two-component 'split-drive' system in the yellow fever mosquito *Aedes aegypti* and provide a proof-of-principle example of drive activity at a single genomic locus. They analyze U6 pol-III regulatory sequences suitable for driving sgRNA expression from plasmids or transgenic strains, and identify combinations of transgenic U6-sgRNA lines and existing Cas9 expression lines that generate somatic and germline mutagenesis and support super-Mendelian inheritance of the U6-sgRNA drive component at the *white* gene on chromosome 1. They extrapolate from the inheritance and somatic mutagenesis data generated when targeting the *white* gene and perform modelling to suggest that strains exhibiting these properties might be effective tools for mosquito control while retaining self-limiting properties that might be desirable from a regulatory standpoint. Overall, the characterization of U6 regulatory sequences that function to drive sgRNA expression is a significant advance for the field. However, based on these results, it seems premature to extrapolate from results seen at a single genomic locus to generalizations regarding the relative merits of different drive systems in *Aedes aegypti*. In particular, the authors make the point that alterations in the timing, spatial patterns, and levels of expression of the gene drive components induce significant variability in somatic mutagenesis and drive phenotypes, meaning that any attempt to adapt the system described here to a novel genomic locus (or even a system utilizing the same genomic locus but with a different drive cargo) may require re-optimization of each genetic element comprising the drive. Thus, as a whole, this work represents a successful proof-of-principle of a genetically encoded CRISPR/Cas9 gene drive in *Ae. aegypti*, but it is unclear how generalizable these optimizations and results will be to future control efforts.

Specific issues to be addressed include:

1) The conservation analysis of the *white* gene sgRNA site is based almost entirely on mosquitoes from the Americas. The authors should either temper the conclusion that this sgRNA recognition site is likely 'fixed in wild populations' and/or do a more thorough analysis of existing sequencing data from additional geographical regions. In addition, the sequencing is reported to be to 'approximate depth of 10x' which raise concerns about the accuracy and sensitivity of variant calling as it is unclear how many reads map to the desired locus – the authors should instead report the coverage of the *white* region for each individual and exclude any samples for which there is not sufficient read depth at this site to confidently call variants.

2) Only one line was established for the U6[a-d] lines in the experiment shown in Figure 1C, raising a significant likelihood of position-dependent effects confounding interpretation of the relative efficacy of these four promoters. The results seen with U6d-sgRNA appear to be an indicator of this, showing efficacy when injected as plasmid but not as a transgenic cross. If it is not feasible to test additional insertions, the authors should acknowledge this limitation of this particular experiment and tone down their interpretation presented in subsection “Development of a binary CRISPR approach”.

3) The sex-linkage of transgenic components introduced to the *white* gene is not unexpected. The authors should cite the numerous studies (e.g. Fontaine et al., 2017) showing that there is an extensive region (> 100Mb in some strains) of markedly reduced recombination on chromosome 1.

4) The modelling in Figure 4 suggests that the split drive would outperform a linked system based on the discovery of resistance alleles – however, if more work was done to restrict Cas9 to the germline as in recently reported Anopheline gene drives (using 3' regulatory sequences that likely act to restrict Cas9 expression to the germline and reduce somatic leakiness, for example), that might not be an issue. The authors should discuss whether the development of resistance alleles is in fact a desirable trait, and if so, they should justify this when discussing the modelling results comparing linked and split drives.

5) In the molecular genotyping of the *w^GDe^* insertions in Figure 2—figure supplement 2B, it looks as if the band for the right-side integration of the U6d construct is larger than [a-c]. Is this expected? Did Sanger sequencing reveal an imperfect integration that might explain why U6d was no effective in this context?

6) One thing that is consistently missing in most of the work published on gene-drive and gene-editing is the lack of fitness-cost studies. It would very informative to perform a dedicated, in-depth fitness cost studies to validate the mathematical model presented here and elsewhere.

---

## [Author Response]

Specific issues to be addressed include:1) The conservation analysis of the white gene sgRNA site is based almost entirely on mosquitoes from the Americas. The authors should either temper the conclusion that this sgRNA recognition site is likely 'fixed in wild populations' and/or do a more thorough analysis of existing sequencing data from additional geographical regions. In addition, the sequencing is reported to be to 'approximate depth of 10x' which raise concerns about the accuracy and sensitivity of variant calling as it is unclear how many reads map to the desired locus – the authors should instead report the coverage of the white region for each individual and exclude any samples for which there is not sufficient read depth at this site to confidently call variants.

We thank the reviewers for pointing this out. Given that our conservation analysis was previously almost entirely of the Americas, we have changed the ‘fixed’ to ‘likely conserved’ (see quote below) and have included an additional 6 samples from one location in Puerto Rico (Guaynabo), one location in Mexico (Cuernavaca) and also 3 locations in South Africa (Pretoria, Rooipoort, and Shingwedzi) to expand this analysis. We have also provided the depth information on the target site for each sample (Figure 1—source data 1) and removed samples with low coverage (<4X). Median mean depth of the target site coverage was 10.6X.

“Additionally, we checked the polymorphism data for *Ae. aegypti* available on https://www.vectorbase.org (accessed 1/2/19) and found no published SNP’s in the region of the *white* target site. We therefore consider the target site likely conserved in natural populations.”

2) Only one line was established for the U6[a-d] lines in the experiment shown in Figure 1C, raising a significant likelihood of position-dependent effects confounding interpretation of the relative efficacy of these four promoters. The results seen with U6d-sgRNA appear to be an indicator of this, showing efficacy when injected as plasmid but not as a transgenic cross. If it is not feasible to test additional insertions, the authors should acknowledge this limitation of this particular experiment and tone down their interpretation presented in subsection “Development of a binary CRISPR approach”.

We have tempered this statement and now indicate that the variation seen in mosaicism and knock out frequencies between U6 promoters could be due to variation in promoter expression or due to position dependent effects.

3) The sex-linkage of transgenic components introduced to the white gene is not unexpected. The authors should cite the numerous studies (e.g. Fontaine et al., 2017) showing that there is an extensive region (> 100Mb in some strains) of markedly reduced recombination on chromosome 1.

We have amended the expectation of no sex linkage associated with the *white* target site, as the reviewer indicates this should have been expected from previous literature. A statement is also included to highlight the low recombination rate at this loci further preserving the sex linkage, which is accompanied by appropriate citations of this work.

4) The modelling in Figure 4 suggests that the split drive would outperform a linked system based on the discovery of resistance alleles – however, if more work was done to restrict Cas9 to the germline as in recently reported Anopheline gene drives (using 3' regulatory sequences that likely act to restrict Cas9 expression to the germline and reduce somatic leakiness, for example), that might not be an issue. The authors should discuss whether the development of resistance alleles is in fact a desirable trait, and if so, they should justify this when discussing the modelling results comparing linked and split drives.

We have added the following text to the relevant section of the Results to address this reviewer’s point: “The number of in-frame resistant alleles generated could be reduced by carrying out linked system releases of a similar magnitude to those for the split drive system, or by using a linked system with a higher accurate homing rate, perhaps achievable by restricting Cas9 expression to the germline (Hammond et al., 2018; Kandul et al., 2019).” As discussed in the Results section, the development of resistant alleles does not prevent the linked system from spreading to significant levels in the neighboring population. It also reduces the efficacy of the drive in terms of spreading effector genes to high levels in the population, and so it would seem there is no benefit to them in this example.

5) In the molecular genotyping of the w^GDe^ insertions in Figure 2—figure supplement 2B, it looks as if the band for the right-side integration of the U6d construct is larger than [a-c]. Is this expected? Did Sanger sequencing reveal an imperfect integration that might explain why U6d was no effective in this context?

The Sanger sequencing indicated an extra of 276 bp insertion of the right homology arm location of the U6d line via an imperfect integration, this is not expected and may or may not influence the HDR efficacy. We have added this unexpected result into our Results section.

6) One thing that is consistently missing in most of the work published on gene-drive and gene-editing is the lack of fitness-cost studies. It would very informative to perform a dedicated, in-depth fitness cost studies to validate the mathematical model presented here and elsewhere.

We thank the reviewer for this comment and we agree and have therefore included additional fitness data in a table (Supplementary file 7A). This data includes male and female: fecundity, egg hatch rate, larval to pupal development days, mating competitiveness and median survival comparing wildtype to gRNA only, Cas9 only and split drive. This is more fitness data than is usually accompanied with these studies. Moreover, it should be noted that fitness is a difficult parameter to measure and would require a level of effort beyond the current scope of the paper.